# New Avoparcin-like Molecules from the Avoparcin Producer *Amycolatopsis coloradensis* ATCC 53629

Gina Ramoni [1], Carmine Capozzoli [1], Adriana Bava [1], Francesca Foschi [2], Gianluigi Broggini [2] and Fabrizio Beltrametti [1,*]

1   BioC-CheM Solutions Srl, Via R. Lepetit, 34, 21040 Gerenzano (VA), Italy;
    gramoni@bioc-chemsolutions.com (G.R.); ccapozzoli@bioc-chemsolutions.com (C.C.);
    abava@bioc-chemsolutions.com (A.B.)
2   Science and High Technology Department, Insubria University, Via Valleggio, 9, 22100 Como (CO), Italy;
    francesca.foschi@uninsubria.it (F.F.); gianluigi.broggini@uninsubria.it (G.B.)
*   Correspondence: fbeltrametti@bioc-chemsolutions.com

**Abstract:** *Amycolatopsis coloradensis* ATCC 53629 is the producer of the glycopeptide antibiotic avoparcin. While setting up the production of the avoparcin complex, in view of its use as analytical standard, we uncovered the production of a to-date not described ristosamynil-avoparcin. Ristosamynil-avoparcin is produced together with α- and β-avoparcin (overall indicated as the avoparcin complex). Selection of one high producer morphological variant within the *A. coloradensis* population, together with the use of a new fermentation medium, allowed to increase productivity of the avoparcin complex up to 9 g/L in flask fermentations. The selected high producer displayed a non-spore forming phenotype. All the selected phenotypes, as well as the original unselected population, displayed invariably the ability to produce a complex rich in ristosamynil-avoparcin. This suggested that the original strain deposited was not conforming to the description or that long term storage of the lyovials has selected mutants from the original population.

**Keywords:** avoparcin; ristosamine; ristosamynil-avoparcin; *Amycolatopsis coloradensis*; glycopeptide antibiotics; strain improvement

## 1. Introduction

Actinomycetes are important producers of bioactive secondary metabolites of industrial interest which include antibacterials, anticancer, antifungals, immunosuppressants, and herbicides. Despite the description of over 10,000 bioactive secondary metabolites produced by Actinomycetes (representing more than 45% of all bioactive microbial metabolites discovered) [1], most of the potential of each single producer actinomycete remains hidden within their genomes. Indeed, genome sequencing revealed that actinomycetes, and in general microbes with large genomes, have the genetic information to potentially produce up to 30–50 secondary metabolites [2]. The evolutionary meaning of this "biosynthetic dark matter" is currently being explored in relation to the division of labor which, besides higher organisms, occurs in actinomycetes and in other multicellular or colony-forming microorganisms. In actinomycetes, which are typically solid-substrate-growing microorganisms displaying a life cycle characterized by a complex morphological and biochemical differentiation [3], the genetic potential could be activated/deactivated in a part of the microbial population according to function, spatial position in the growing colony, and temporal requirements [4]. Of particular importance is the expression of the hidden genomic potential by the vegetative hyphae (or substrate mycelium). The substrate mycelium (in some cases defined as the "sterile caste") has indeed the task to surround, nourish, and protect spore formation. The possibility for the substrate mycelium to diversify the production of secondary metabolites could offer selective advantages over competing microorganisms [5]. The potential for production of different secondary metabolites can be activated by the

mere expression of specific genes and by point mutation and/or irreversible rearrangements of the genome [6]. Point mutation and irreversible rearrangements of the genome could result in lineages which display specific characteristics including the modulation of production of secondary metabolites (high or low producers) or the synthesis of different secondary metabolites [4]. These characters are not necessarily inherited by spores and can be preserved artificially in laboratory or production plants and exploited for the industrial production of metabolites.

Actinomycetes of the genus *Amycolatopsis* have a high genetic potential to produce secondary metabolites and bioactive molecules [7]. Strains belonging to this genus are in fact the producers of some of the most widely used antibiotics: rifampicin A, vancomycin, balhimycin, nogabecin, and ristocetin produced, respectively, by *A. mediterranei* or *A. rifamycinica*, *A. orientalis*, *A. balhimycina*, *A. keratiniphila*, and *A. lurida* [8]. Among these, the most important are rifampicin and vancomycin, used as broad-spectrum antibiotics, respectively, for the treatment of tuberculosis and as a last line of defense for the treatment of infections from multi-resistant Gram-positive microorganisms [8].

The interest in *A. coloradensis* ATCC 53629 is linked to the production of the glycopeptide antibiotic avoparcin, also known as LL-AV290 [9,10]. Avoparcin administration was performed in sub-therapeutic doses for prolonged periods of time in intensive farming to prevent disease (prophylaxis) and promote growth [11]. Due to the misuse of this antibiotic, avoparcin is commonly considered as one of the causes of the onset of glycopeptide resistance in pathogenic bacteria (in particular Glycopeptide-Resistant *Enterococcus*, GRE) [12,13]. For this reason, avoparcin was banned in 1995 but its illegal use is still a health issue and needs monitoring [14]. Unfortunately, monitoring is hampered by the difficulty of finding on the market analytical standards of $\alpha$-avoparcin and $\beta$-avoparcin, the major components of the complex product. With the purpose of obtaining an analytical standard, we isolated morphological variants, we developed a fermentation medium and a purification method for the quantitative production of avoparcin analytical standards. Surprisingly, while producing avoparcin, we were able to identify new avoparcin-like glycopeptides produced by the avoparcin producer *A. coloradensis* ATCC 53629. The production and characterization of the new molecules is described in this work.

## 2. Materials and Methods

### 2.1. Strains, Cultivation Conditions and Identification of Morphological Variants

The original avoparcin producer strain *Amycolatopsis coloradensis* ATCC 53629 was purchased from the DSMZ strain collection in 2018. The strain was routinely preserved as a frozen master cell bank (MCB). Working cell banks (WCBs) were prepared from agar plates or slants of Medium V0.1 [15] originating from the MCB as already described for other actinomycetes [16]. Cryo-vials of 1.0 mL were stored at –80 °C for up to two years without influence on the avoparcin and ristosamynil-avoparcin production. Mycelium for colony isolation was prepared as already described [17,18]. Colonies were selected based on the different morphology, with the support of a Carl Zeiss Stemi SV 6 stereomicroscope (Carl Zeiss, Jena, Germany). Agar media for selection were ISP2 (composition in g/L: Yeast extract (Costantino & C SpA, Favria, Italy) 4.0 g; Malt extract (Costantino & C SpA, Favria, Italy) 10.0; Dextrose (A.D.E.A Srl, Busto Arsizio, Italy) 4.0; Agar (HiMedia, Schenzhen, China) 20.0; pH 7.2 with NaOH/HCl [(Carlo Erba Reagents Srl, Cornaredo, Italy)), BTT-P (composition in g/L: Yeast extract (Costantino & C SpA, Favria, Italy) 1.0; Meat extract (Costantino & C SpA, Favria, Italy) 1.0; Casein hydrolysate (Costantino & C SpA, Favria, Italy) 2.0; Dextrose (A.D.E.A Srl, Busto Arsizio, Italy) 10; Agar (HiMedia, Schenzhen, China) 15; pH 7.2 with $K_2HPO_4$ (Carlo Erba Reagents Srl, Cornaredo, Italy)) and BTT-0.25 (composition as for BTT-P but diluted $\frac{1}{4}$—besides agar—with ultrapure water).

Fermentation for avoparcin and ristosamynil-avoparcin production was performed as follows. Cryo-vials of the WCB were thawed at room temperature, and 2 mL were used to inoculate 50 mL of Medium 3–9 [19] in 500 mL baffled flasks and grown for 48 h on a rotary shaker at 240 rpm and 28 °C. Medium 3–9 components were in g/L:

Cane Molasses (Kivinat, Saint-Marcel-Lès-Valence, France) 20; N-Z amine type A (Merck KGaA, Darmstadt, Germany) 10; pH of the medium prior to autoclaving was adjusted to 7.5 with NaOH/HCl (Merck KGaA, Darmstadt, Germany). Fermentation was started by adding a 10% (vol/vol) inoculum from the vegetative medium flask into 50 mL of BCS360 production medium in 500 mL flasks. BCS360 components were in g/L: bacto-yeast extract (Costantino & C SpA, Favria, Italy) 9; soybean flour (Costantino & C SpA, Favria, Italy) 10; dextrose (A.D.E.A Srl, Busto Arsizio, Italy) 15; soluble starch (Carlo Erba Reagents Srl, Cornaredo, Italy) 10; $CaCO_3$ (Merck KGaA, Darmstadt, Germany) 4; pH of the medium prior to autoclaving was adjusted to 7.5 with NaOH. Foam production was controlled by dropping antifoam (Hodag; Vantage, Deerfield, IL, USA) whenever required. We used 0.5 mL culture samples for daily monitoring of pH, glucose, and glycerol consumption, and for the production of avoparcin and ristosamynil-avoparcin. A total of 10 mL of culture was collected from a parallel set of flasks for Packed Mycelium Volume (PMV %).

For glycerol and glucose analysis, samples were mixed with 10% (*v/v*) of a 35% $HClO_4$ (*v/v*) solution, incubated at $-20$ °C for 10 min and then neutralized with 7 M KOH (*w/v*) (both reagents from Carlo Erba Reagents Srl, Cornaredo, Italy). Monitoring of glycerol and glucose was performed by High Performance Liquid Chromatography (HPLC) on a Aminex HPX-87H cationic exchange column (300 × 7.8 mm) (BioRad, Hercules, CA, USA) eluted at 30 °C at a flow rate of 0.6 mL/min with 5 mM $H_2SO_4$ in water (Carlo Erba Reagents Srl, Cornaredo, Italy) for 20 min. Chromatography was performed with an HPLC Agilent 1260 Infinity system (Agilent Technologies Inc., Santa Clara, CA, USA) equipped with a Refraction Index Detector (RID) and with an UV detector set at 210 nm. Glucose and glycerol were quantified by comparison with real standards (purity of >95%) (Merck KGaA, Darmstadt, Germany).

### 2.2. Avoparcin Complex Extraction, Analysis and Purification

The avoparcin complex of molecules was extracted by mixing 1 volume of whole fermentation broth and 1 volume of distilled water. Samples were then vortexed 1 min and finally centrifuged (16,000× *g* for 5 min). The glycopeptide-containing supernatant was filtered through a Durapore membrane filter (0.22 μm) (Merck Millipore, Burlington, MA, USA). Glycopeptide production was estimated by High Performance Liquid Chromatography (HPLC) performed on a Inertsil ODS-3 column (250 × 4.6 mm, 5 μm) (GL Sciences, Tokyo, Japan) eluted at 40 °C at a flow rate of 1.4 mL/min with a 35-min gradient as follows: 0 min = 3% Phase B; 30 min = 50% Phase B; 30.1 min = 95% Phase B; 35 min = 95% Phase B; 35.1 min = 97% Phase B. Phase A was 0.3% HCOOH (Merck KGaA, Darmstadt, Germany)–water (Carlo Erba Reagents Srl, Cornaredo, Italy), and Phase B was 0.3% HCOOH–Methanol (Carlo Erba Reagents Srl, Cornaredo, Italy) mixture. Chromatography was performed with an HPLC diode array Agilent 1260 Infinity system (Agilent Technologies Inc., Santa Clara, CA, USA) with detection at 280 nm. DAD spectra were acquired between 190 and 400 nm and the acquired spectra were compared with those described in [20]. Avoparcin standard produced at BioC-CheM Solutions (purity of >95%) was used as an internal standard.

Assignment of the structure of the avoparcin complex component was obtained by use of Tandem Liquid Chromatography–Mass Spectroscopy (LC/MS) experiments operating in positive-ion mode (ESI), and NMR studies. The mass spectrometric detection was realized using a single quadrupole LC/MSD system (Agilent Technologies Inc., Santa Clara, CA, USA) operating in positive-ion mode (ESI). $^1$H and $^{13}$C NMR spectra were recorded on a Bruker (Billerica, MA, USA) Avance 400 (400 and 100.6 MHz, respectively); chemical shifts are indicated in parts per million (ppm) downfield from SiMe$_4$, with residual proton (($CH_3)_2$SO = 2.50 ppm, HOD = 4.80 ppm]) and carbon (($CD_3)_2$SO = 40.45 ppm) solvent resonances as internal reference. Protons and carbon investigations were achieved by $^{13}$C attached proton test (APT), $^1$H$-^1$H correlation spectroscopy (COSY), and $^1$H$-^{13}$C heteronuclear correlation experiments.

Quantitative recovery and purification of the glycopeptide antibiotic from the fermentation broth was performed by use of the affinity resin Sepharose-D-alanyl- D-alanine [21]. For the processing of 2 L of fermentation broth containing ca. 15 g of avoparcin complex and approximately 700 mL of resin were loaded on a 7 cm wide glass column. The purification procedure was as follows. Two liters of whole fermentation broth were centrifuged to separate the supernatant from the mycelium (residual supernatant volume 1700 mL). Mycelium was washed with 700 mL of deionized water to recover residual avoparcin. Approximately 2.4 L of avoparcin enriched broth were then corrected at pH 7.5 with HCl 1 M (Carlo Erba Reagents Srl, Cornaredo, Italy) and loaded on the column at a speed of 10 mL/min. After loading, the column was washed with 2 L of deionized water, 2 L of 0.1 M Acetic Acid (Carlo Erba Reagents Srl, Cornaredo, Italy) and 2 L of 0.1 M $NH_4OH$ (Carlo Erba Reagents Srl, Cornaredo, Italy). Avoparcin was then eluted with a linear gradient from 0.3 M to 0.5 M $NH_4OH$. The avoparcin enriched fractions were then pooled, corrected at pH 4 with 5% $H_2SO_4$ (Carlo Erba Reagents Srl, Cornaredo, Italy) and avoparcin was precipitated by dropping the above solution into 4 volumes of Acetone (Carlo Erba Reagents Srl, Cornaredo, Italy).

## 3. Results

### 3.1. Identification of a Heterogeneous Population in A. coloradensis ATCC 53629

It is known that the chromosome of the model actinomycetes belonging to the genus *Streptomyces* undergoes large spontaneous deletions at rates higher than 0.1% of seeded spores [22]. Tandem amplifications of specific chromosomal regions and associated deletions occur, and RecA seems to be involved in the control of this genetic instability [22]. These genome modifications could result in modification in the productivity of secondary metabolites and even in the production of new metabolites [4]. A similar molecular mechanism could be the rule in other actinomycetes as increasing evidence of phenotypic variation are piling up [17]. Based on the above statements, we started our study with the analysis of the morphology of the avoparcin producer strain. From the seeding of the original lyovial purchased from the DSMZ strain collection on three different agar media (ISP2, BTT-P and BTT-0.25), we were able to distinguish 12 distinct colony morphologies (identified as C1, C6, C8, C10, C14, C17, C18, C22, C23, C24, C29, and C30 (Figure 1 and Table S1). The preliminary analysis of these morphologies revealed that all were able to produce different amounts (ranging from ca 1 to ca 9 g/L) of the avoparcin complex with C14 being the highest producer (used in the subsequent experiments). The isolated morphologies were, in general, characterized by a limited production of aerial mycelium and spores. *A. coloradensis* C14 typical colony was around 2.5 mm in diameter, with a fluffy white/light yellow aerial mycelium and no evidence of spores. The colony was typically protruding in the center and breaks appeared with aging.

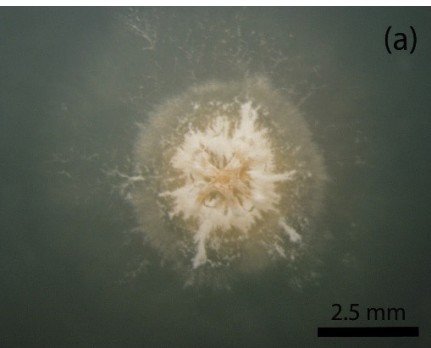 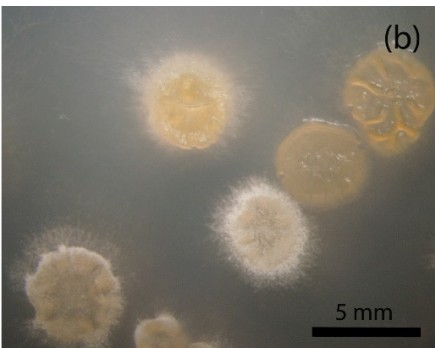 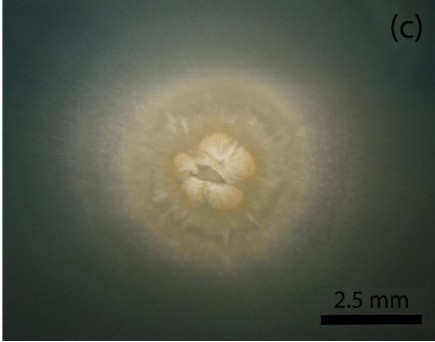

**Figure 1.** (**a**) Typical morphology of *A. coloradensis* growing on Medium V 0.1, (**b**) different morphologies observed on agar medium, and (**c**) high producer *A. coloradensis* C14 grown on Medium V 0.1 agar.

Surprisingly, all the variants were able to produce an alternative avoparcin complex composed mainly of α-avoparcin, β-avoparcin, and their respective ristosamylated analogues (see below).

### 3.2. Fermentation of A. coloradensis ATCC 53629 for the Production of Avoparcin

The screening of the best producer medium for avoparcin, was performed with the database BCSMedDat, a proprietary database of industrial media for the production of secondary metabolites of interest. The selected medium, BCS360, was characterized by a neutral pH and by the presence of both glucose and glycerol as carbon sources. The fermentation of *A. coloradensis* C14 occurred with the concomitant utilization of glucose and glycerol. The onset of avoparcin production initiated roughly with glucose depletion and proceeded till glycerol decreased below 10 g/L. Production reached values above 9 g/L in 120–144 h of fermentation and biomass production reached a maximum PMV level of 25% with a SD of ±5%. The pH during fermentation was in the range 7.2–7.6 (Figure 2).

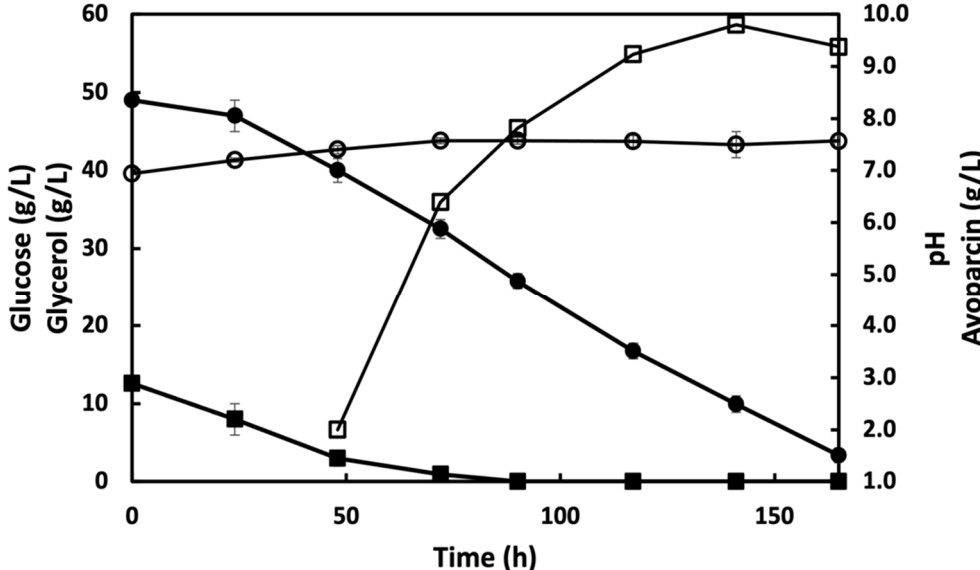

**Figure 2.** Time course of avoparcin complex production by *A. coloradensis* C14 cultivated in BCS360 medium. Glucose consumption (filled squares), Glycerol consumption (filled circles), pH (circles), and avoparcin complex production (squares) were monitored every 24 h. Avoparcin complex includes α- and β-avoparcin, and α- and β-ristosamynil-avoparcin.

### 3.3. Purification and Chemical Characterization of the Avoparcin Complex

The level of production obtained through the combination of the high producer clone C14 and of the medium BCS360, allowed to obtain a significant amount of avoparcin complex directly from flask fermentation. A set of 10 parallel flasks (100 mL each), was run and harvested at the productivity peak. The product was then purified by use of agarose D-Ala-D-Ala resin [21], as described in the section Materials and Methods. From this experimentation, a quantity of avoparcin (9.27 g from 2 L of broth culture) was obtained in a solid form. This amount was sufficient for a fine characterization of the molecule and to produce a certificate of analysis. The purified product was analyzed by NMR and LC-MS as described in the section Materials and Methods. The avoparcin complex consisted mainly of the two structurally related glycopeptides: α-avoparcin and β-avoparcin as minor and major components, respectively (Figure 3) [20,23–28]. As reported in the literature, together with α- and β-avoparcin various types of epimers or hydrolysis products at the sugar level of the avoparcin aglycones have been detected and identified [20,23–28]. On the contrary, avoparcin fermentation products bearing additional sugar units are up to date not described. The development of the HPLC analytical method allowed us to define the characteristics of the avoparcin fermentation complex (Figure 3). Together with the

α- and β-avoparcin signals (retention time (rt) α-avoparcin: 8.050 min; rt β-avoparcin: 8.813 min) the HPLC trace revealed the presence of two additional peaks (rt Unknown (U) 1: 8.289 min; rt U 2: 9.099 min, Figure 3). Interestingly, the UV absorption spectra of U 1 and U 2 signals showed the same profile as that observed for α- and β-avoparcin, indicating a close relation between the chemical structures of compounds U 1 and U 2 and α- and β-avoparcin. High-Resolution Liquid Chromatography-Tandem Mass spectrometry (LC-MS) experiments were also performed to gain additional structural information. The full-scan LC-MS (ESI) spectra of the fermentation complex confirmed the presence of the protonated ions at $[M + 2H]^{2+}$ of α- ($m/z$ 956) and β-Avoparcin ($m/z$ 973) [20,29]. On the other hand, the LC-MS response related to U1 and U2 could not be traced back to any literature data. In particular, the MS spectra of compounds U1 and U2 are characterized by the presence of cluster ions at $m/z$ 1022 and 1039. The fragmentation of these latter ions gave rise to the ions at $m/z$ 956 and 973, corresponding to those of α- and β-Avoparcin. Thus, signals at $m/z$ 1022 and 1039 are consistent to an increase of 66 to the α- and β-Avoparcin $m/z$ values. It is worth of note that variation of 66 in the electrospray ionization experiments of Avoparcin aglicones is correlated to a variation at the ristosamine sugar [20,23]. These data suggested that the U1 and U2 formation could arise from an additional ristosamine units in α- and β-avoparcin, respectively. Thus, unknown glicopeptides U1 and U2 were identified as a α-ristosamynil-avoparcin and β-ristosamynil-avoparcin. NMR analysis displayed that the $^1$H chemical shift related to the avoparcin fermentation complex showed 4 signals in the field correspondent to the aromatic proton in D6 position (6.55–6.40 ppm, Figures 4 and 5). Hence, we assumed that functionalization occurs at level of the D aryl ring. NMR analysis allowed therefore to hypothesize two potential sites of the molecule which could be attached to ristosamine (Figure 4).

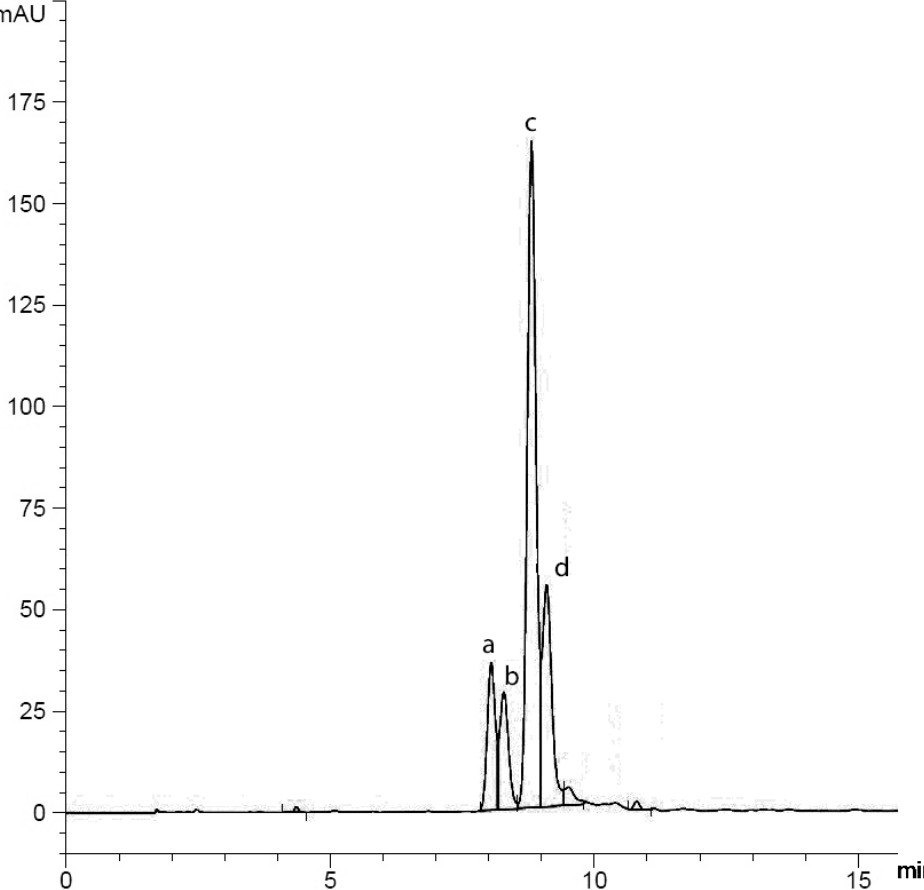

**Figure 3.** HPLC profile of the agarose D-Ala-D-Ala purified avoparcin complex. (a) α-avoparcin, (c) β-avoparcin (b) α-ristosamynil-avoparcin, and (d) β-ristosamynil-avoparcin.

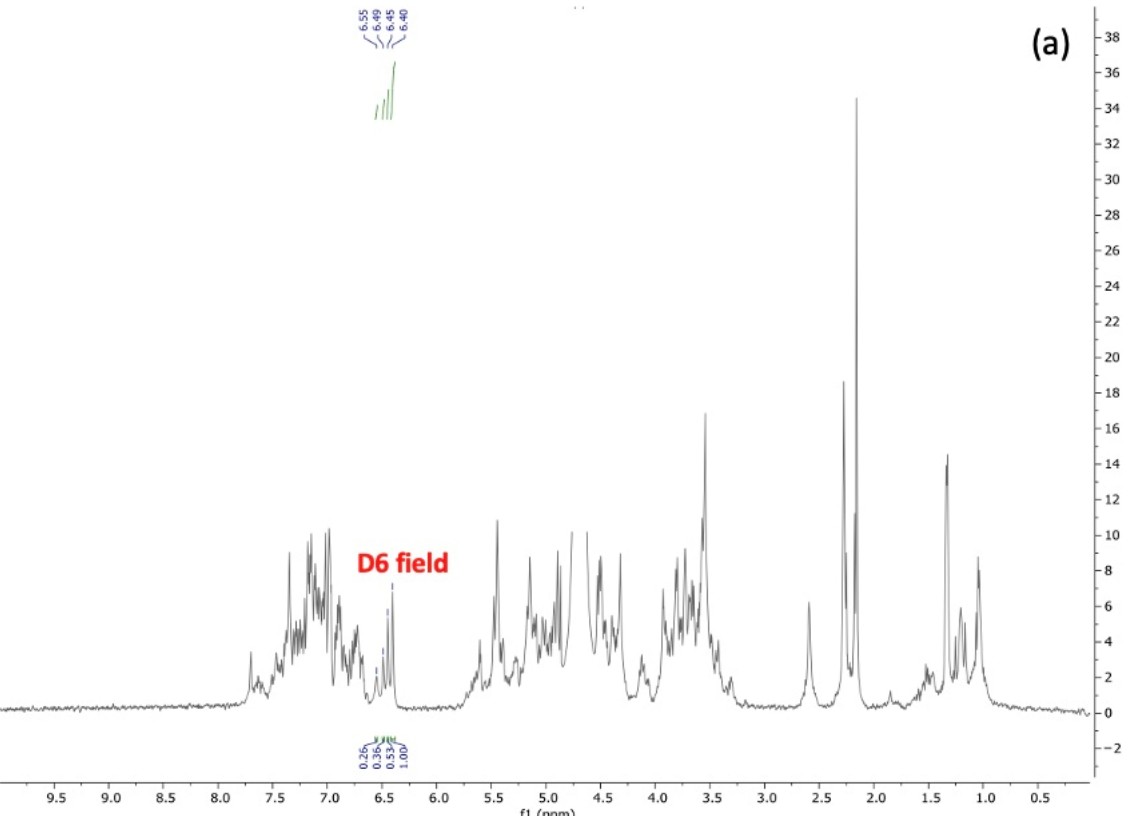

**Figure 4.** Chemical structure of avoparcin possible sites of attachment of the second ristosamine residue giving rise to the respective ristosamynil-avoparcin are shown by arrows. Functionalization occurs at the level of the D aryl ring.

**Figure 5.** *Cont.*

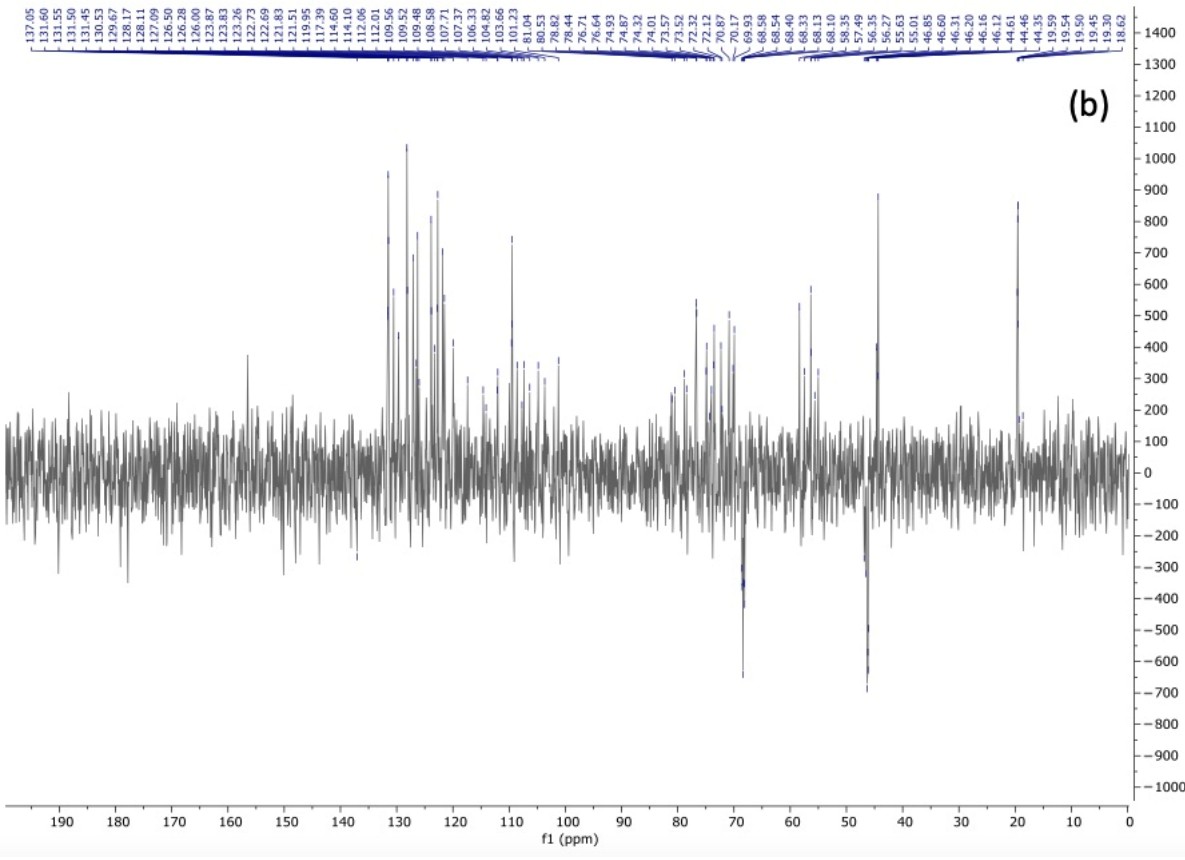

**Figure 5.** $^1$H NMR Spectra of the avoparcin fermentation complex (D$_2$O; 400 MHZ—significant signals are integrated) (the D aryl ring signals are shown) (**a**), and $^{13}$C NMR Spectra of the avoparcin fermentation complex (D$_2$O; 100 MHZ) (**b**).

Despite the above results, due to the low amount of α- and β-ristosamynil-avoparcin together with their high molecular weight (α-ristosamynil-avoparcin relative molecular mass: 2038; β-ristosamynil-avoparcin: 2072) the exact regiochemistry of α- and β-ristosamynil-avoparcin still remains under discussion.

## 4. Discussion

Examples of bacterial differentiation that result in morphological change have been known for decades. In *A. coloradensis*, we observed that the population originating from the lyovial directly purchased from the DSMZ strain collection (Braunschweig, Germany), displayed a high number of morphological variants when seeded on different agar media. Different avoparcin complex productivities were also evidenced and, noteworthy, the isolated morphological variants, as well as the most represented population (which we considered as the "wild type") invariably produced a modified avoparcin complex rich in the not yet described glycopeptide ristosamynil-avoparcin. It was reported that the *Amycolatopsis* genus displays relevant genomic plasticity hotspots (defined as quasi-core regions to distinguish them from core regions in which the plasticity is reduced) [30]. This genomic plasticity was elegantly connected to the production of antibiotics, by the analysis performed by Kumari and co-workers [7]. The authors analyzed twelve strains of rifamycin producing *A. mediterranei.* The strains were independent isolates or mutants with the ability to produce different rifamycins (rifamycin B, SV, P, Q, R, U, W) and it was shown that the identification of new lineages producing rifamycin analogs and/or overproducing a single rifamycin factor occurred with relative ease thus supporting the plasticity of the genome of the strain. The analysis of the genome of *A. mediterranei* has also shown the presence of more than 30 gene clusters for the biosynthesis of secondary metabolites [31] while in *A. orientalis*

HCCB10007 (the industrial vancomycin producer strain) twenty-six secondary metabolite biosynthetic gene clusters were predicted [30]. To date, 67 *Amycolatopsis* genomes, including the one from *A. coloradensis* ATCC 53629, have been completely sequenced and they range in size from 5.62 Mbp in *A. granulosa* to 10.94 Mbp for *A. anabasis* (source NCBI datasets: https://www.ncbi.nlm.nih.gov/datasets/, accessed on 17 January 2022). The complete genome sequence of *A. coloradensis* ATCC 53629 has a notable size of 9.05 Mbp and contains 8364 genes (NCBI Reference Sequence: NZ_MQUQ00000000.1, accessed on 17 January 2022). Although details for *A. coloradensis* potential to produce secondary metabolites was not studied in detail, data collected for *A. orientalis*, *A lurida*, and *A. mediterranei*, show the presence of 25 to 36 secondary metabolite gene clusters indicating a noteworthy hidden potential for the synthesis of new metabolites [32]. In conclusion, *Amycolatopsis* possess several biosynthetic gene clusters (BGCs) that can be of biological importance, and which can arise during standard programs of strain improvement and strain maintenance. It was therefore not surprising to find a new avoparcin complex during our work. What was surprising, was rather that, upon testing of different fermentation media (including the one described in the original patent from Murray et al. [19]), we were able to invariably produce a complex of molecules with a high amount (up to 40%) of ristosamynil-avoparcin factors. We have reasoned that this result could have different potential origins which will need to be investigated in detail. Among our hypothesis for this effect, the lyophilization process could have been a selection system which drastically reduced the number of regenerating genomes eventually contained within the stored hyphae. In example, survival rates after the drying of *Escherichia coli*, *Pseudomonas putida*, *Serratia marcescens*, and *Alcaligenes faecalis*, decreased for the first 5 years upon lyophilization and then stabilized to around 10% thereafter [33]. We can therefore argue that the surviving 10% of the population was somehow selected by the storage. It was also reported that freeze drying influenced the antibiotic resistance of the preserved strain [34]. Therefore, the selection of mutants in the lyovials of *A. coloradensis* could arise from the long-term storage under suboptimal conditions.

In industrial programs of strain and fermentation improvement, the so-called strain maintenance protocol is routinely applied, based on the selection of the best clonal populations growing on agar media to then scale them up in large-scale, submerged, culture-based fermentations. Selection at this stage is often based on phenotypic traits (colony morphology, color, or simply increased productivity) that are basically applied blindly [28]. In addition, bacterial populations contain phenotypic cell variants that lack morphological change. In *A. coloradensis*, the classical isolation of different morphologies was fruitful in giving rise to potentially industrial producers with high incidence. A different colony morphology in plate was associated with the increased antibiotic productivity, confirming that the empirical practice based on colony morphology observation still represents a valid tool, especially if applied to the original wild-type isolates, which exhibit some intrinsic heterogeneity. Although recombinant engineering is increasingly attracting industrial interest, allowing specific genetic modifications in the wild-type background, a preliminary analysis of the wild-type population is recommended to avoid misinterpretation of the results.

From the practical point of view, the isolation of morphological variants of *A. coloradensis,* allowed the identification of high producers with relative ease, indicating that the variability associated with the *Amycolatopsis* chromosome could be an evolutive strategy as already described for *Streptomyces* and *Actinoplanes*. It is worth of note that at least 20% of the morphological variants which we have identified in the population, were unable or seriously hampered in the production of aerial mycelium and spores. This suggests that a general concept of division of labor could apply also to *Amycolatopsis* as already reported for *Streptomyces* and *Actinoplanes* and that the sterile caste could play a crucial role in specific aspects of the *Amiycolatopsis* growth.

In the light of the continuous urge in new, cutting-edge antibiotics [35], it will be of absolute interest to test the antimicrobial activity of the newly identified molecules and also to test the possibility to semi-synthetically improve their clinical characteristics. From

the industrial point of view, the production of more than 9 g/L of the avoparcin complex will allow a cheap production process.

## 5. Conclusions

*A. coloradensis* ATCC 53629 clonal populations selected based on their colony morphology produced different amounts of avoparcin in fermentation broth. The different morphologies, with no exclusion, were invariably able to produce a complex of molecules which is consistently different from the one described in the literature [26]. The sum of the new identified factors α- and β-ristosamynil-avoparcin represents indeed ca. 30% of the avoparcin complex upon purification with affinity resins. This is the first description of ristosamynil-avoparcin and this result supports the idea of any single actinomycete as a source of several bio-active molecules which are uncovered whenever suitable conditions arise. Concerning the rise of this peculiar avoparcin complex, we speculated that long term storage in freeze dried condition could have selected recessive mutations which determined the production of the new molecules. This observation also uncovered the risks of a suboptimal storage system which could result in loss of the original phenotypes by microorganisms deposited in strain collections with a consequent loss in biodiversity and, specifically for *A. coloradensis*, the potential presence in the illegal market of alternative and unknown (and mostly undetectable by standard tests) avoparcin products.

**Supplementary Materials:** The following supporting information can be downloaded at: https://www.mdpi.com/article/10.3390/fermentation8020044/s1, Table S1: Description of the morphological variants isolated from *A. coloradensis* ATCC 53629. Each independent clone was cross-checked for purity and stability of the phenotype on different agar media.

**Author Contributions:** Conceptualization, F.B. and A.B.; investigation, G.R., C.C., F.F. and A.B.; methodology, F.B., G.B. and A.B.; supervision, F.B. and C.C.; writing—original draft preparation, F.B., G.B. and A.B.; writing—review and editing, F.B., A.B. and C.C. All authors have read and agreed to the published version of the manuscript.

**Funding:** This research received no external funding.

**Informed Consent Statement:** Not applicable.

**Data Availability Statement:** Data is contained within the article or supplementary material.

**Acknowledgments:** We are grateful to Luca Mellere for technical support with figures and graphics.

**Conflicts of Interest:** The authors declare no conflict of interest.

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
