# Peer review of "New Avoparcin-like Molecules from the Avoparcin Producer Amycolatopsis coloradensis ATCC 53629"

_fermentation, doi:10.3390/fermentation8020044_

Round 1
Reviewer 1 Report
In this manuscript, Ramoni et al. analyzed the morphological phenotype and secondary metabolites of Amycolatopsis coloradensis. They identified a new ristosamynil-avoparcin, produced by a A. coloradensis with a non-spore forming phenotype, could be used as analytical standard in the future. I would say that the new avoparcin-like molecule is important and valuable for controlling misuse of avoparcin. Overall, this is a well-organized and insightful paper. It is easy to read. Here are some suggestions may strengthen this manuscript:
- Line74-79: here needs a conclusion of the manuscript.
- Line177: any reference to support this sentence “RecA seems to be …”.
- Line 185: “Error! Reference source not found”
- Line 199: species name should be italicized
Author Response
Answers to reviewer 1
We thank the reviewer for considering our work of interest for the scientific community and for suggesting improvements.
Below are answers to specific comments
- Line74-79: here needs a conclusion of the manuscript.
We have deleted the part which remained from the instructions to authors. We are sorry for this mistake but formatting is sometimes playing against us….
We have also added a relevant conclusion for the introduction chapter
- Line177: any reference to support this sentence “RecA seems to be …”.
We have added the relevant reference for the statement
- Line 185: “Error! Reference source not found”
We have corrected the mistake by adding the relevant indication of Figure 1
- Line 199: species name should be italicized
We have italicized the species names where missing
Reviewer 3 Report
The manuscript presented by the authors describes the hyper-production of avoparcin-like compounds from a variant of Amycolatopsis coloradensis ATCC 53629 isolated from the commercial source. The study is very preliminary and lacks convincing evidence to support the conclusion. See specific comments below:
- The manuscript is not well written and has revision comments (by the authors). For example, page 2, line 74-79; page 4, line 185-186.
- The critical evidence for determining the compound structures is lacking. NMR (including 1D, and 2D NMR data) should be provided in the manuscript or the supplementary information.
- The authors should perform 16s RNA analysis to reveal the colony identity. Are those colonies Amycolatopsis coloradensis or contamination?
- The production titers by each morphology should be reported.
- Whole genome sequencing should be performed to identify the mutations of strain C14 that confer the hyper-production phenotype.
Author Response
Answers to reviewer 3
We thank the reviewer for stimulating discussion on the topic. The work could be considered preliminary from the point of view of in-depth characterization of the morphological variants of the strain which have been considered. However, due to the relevant discovery of new avoparcin, we consider of fundamental importance to offer ASAP our discovery to the scientific community. Consider also that, as far as the strain is purchased from DSMZ, it is possible that it had already crossed the borders of potential industrial production.
Concerning the specific comments:
- The manuscript is not well written and has revision comments (by the authors). For example, page 2, line 74-79; page 4, line 185-186.
We have corrected the revision comments. The parts evidenced by the reviewer were unfortunately due to formatting issues which were promptly corrected. We also were along the manuscript in order to correct other formatting issues (see the manuscript). Any further suggestion on how to improve the message which we want to deliver with the manuscript will be promptly considered
- The critical evidence for determining the compound structures is lacking. NMR (including 1D, and 2D NMR data) should be provided in the manuscript or the supplementary information.
The requested NMR data have been included in Figure 5 and Figure 4 was modified with more precise indications. We also have added a clarifying sentence at lines 288-292. The discrimination among the two possible sites of attachment of the second ristosamine residue is at present not feasible as it is difficult to efficiently prepare a sufficient amount of the pure compounds for additional NMR analysis
- The authors should perform 16s RNA analysis to reveal the colony identity. Are those colonies Amycolatopsis coloradensis or contamination?
All the lineages (as well as the original strain) were checked by 16s rDNA sequencing. The presence of contaminants was excluded with standard microscopic and in-plate analysis. Furthermore, production of avoparcin occurred (although at different levels) in 100% of the clones. Based on the above evidence, we exclude the presence of contaminants in our experimental work.
- The production titers by each morphology should be reported.
The titers have been reported in Table S1 for enhanced clarity
- Whole genome sequencing should be performed to identify the mutations of strain C14 that confer the hyper-production phenotype.
We agree with the reviewer. However, at the present level of overall knowledge on the strain, all the selected morphologies should be sequenced and a comparative analysis among them should be performed in order to have hints on the mutations involved in productivity. Furthermore, being unknown the nature of the mutations involved (deletions, insertions, inversions or simple point mutations), a precise sequencing method should be used to evidence even point mutations. Actually, we see this as an independent and extended project which we have already started in collaboration with external working groups.
We also have to add that, for this strain, few methods of genetic manipulation are reported. Therefore, unless we are able to identify some mutation which directly involves the avoparcing synthesis cluster (which will not automatically exclude the presence of other unrelated mutations involved in the resulting phenotype), we should first find the suitable method for artificially reproducing the identified mutations in a wild type background as this will eventually prove that the mutation is actually involved in giving the resulting productivity phenotype
Round 2
Reviewer 3 Report
The NMR data showing the modification happens at D aryl ring is preliminary. However, considering the difficulty of getting sufficient amount of pure compounds for additional NMR confirmation, authors' revision in current form is acceptable.